# Overexpression of *TCP5* or Its Dominant Repressor Form, *TCP5-SRDX*, Causes Male Infertility in Arabidopsis

**DOI:** 10.3390/ijms26051813

**Published:** 2025-02-20

**Authors:** Tingting Li, Ping Tian, Xinxin Wang, Mengyao Li, Shuping Xing

**Affiliations:** 1College of Life Science, Shanxi University, Taiyuan 030006, China; ltt100518@163.com (T.L.); 15535359987@163.com (P.T.); w2233914175@163.com (X.W.); 13753723169@163.com (M.L.); 2Institute of Applied Biology, Shanxi University, Taiyuan 030006, China; 3Shanxi Key Laboratory of Nucleic Acid Biopesticides, Shanxi University, Taiyuan 030600, China

**Keywords:** *TCP5* overexpression, *TCP5-SRDX*, anther dehiscence, anther development, gene regulation, plant fertility

## Abstract

TCP transcription factors have long been known to play a crucial role in leaf development, but their significance in reproduction has recently been revealed. TCP5 is a member of class II of the TCP family, which predominantly regulates cell differentiation. This study used overexpression and SRDX fusion to evaluate the role of TCP5 in anther development. *TCP5* overexpression resulted in lower fertility, primarily due to anther non-dehiscence. We also observed reduced lignin accumulation in the anther endothecium. In addition, *TCP5* overexpression resulted in smaller anthers with fewer pollen sacs and pollen due to early-anther defects before meiosis. *TCP5* showed expression in early anthers, including the epidermis, endothecium, middle layer, tapetum, sporogenous cells (pollen mother cells), and vascular bundles. Conversely, during meiosis, the *TCP5* signal was only detected in the tapetum, PMCs, and vascular bundles. The *TCP5* signal disappeared after meiosis, and no signal was observed in mature anthers. Interestingly, the *TCP5-SRDX* transgenic plants were also sterile, at least for the early-arising flowers, if not all of them. *TCP5-SRDX* expression also resulted in undersized anthers with fewer pollen sacs and pollen. However, the lignin accumulation in most of these anthers was comparable to that of the wild type, allowing these anthers to open. The qRT-PCR results revealed that several genes associated with secondary cell wall thickening had altered expression profiles in *TCP5* overexpression transgenics, which supported the non-dehiscent anther phenotype. Furthermore, the expression levels of numerous critical anther genes were down-regulated in both *TCP5* overexpression and *TCP5-SRDX* plants, indicating a comparable anther phenotype in these transgenic plants. These findings not only suggest that an appropriate *TCP5* expression level is essential for anther development and plant fertility, but also improve our understanding of TCP transcription factor functioning in plant male reproduction and contribute information that may allow us to manipulate fertility and breeding in crops.

## 1. Introduction

TEOSINTE BRANCHED1, CYCLOIDEA, and PROLIFERATING CELL FACTORS (TCPs) constitute a family of plant-specific transcription factors that exist in various plants, from green algae to higher plants [1]. All known TCP members share a conserved TCP domain of 59 amino acids, forming a non-canonical basic helix–loop–helix structure responsible for DNA binding, protein–protein interaction, and dimerization [1,2,3]. TCP family members are classified into two classes, I and II, based on the features of the sequences within and outside the TCP domain. Class I members lack 4 amino acids in the basic region but include 6–7 more amino acids in the second helix than class II [1]. In addition, class I TCPs have short sequences flanking the TCP domain, but most class II TCPs possess an arginine-rich region or R domain that may be crucial for protein–protein interaction [4]. Phylogenetically, class I TCPs are closely related, whereas class II TCPs can be further divided into the CYC/TB1 and CIN (CINCINNATA-like) clades [3]. However, it is uncertain which class of *TCP* genes is more ancient, as even basal plant species have two classes of *TCPs*, even though these species only have a few *TCPs* [1]. A total of 13 of the 24 TCP members in the Arabidopsis genome are members of class I, while 11 are members of class II, which is composed of the CYC/TB1 (TCP1, TCP12/BRC2, and TCP18/BRC1) and CIN (TCP2/3/4/10/24 and TCP5/13/17) clades. In the CIN clade, miR319 regulates the *TCP2/3/4/10/24* genes [5].

TCPs act as transcriptional regulators, binding to the promoters of their target genes and regulating their expression. Both class I and II TCPs bind to GC-rich sequences, but their binding preferences differ. Class I TCPs prefer the sequence GTGGGNCC, while class II prefers GTGGNCCC [3,5]. The two TCP classes might share target genes by binding to the same and different cis-elements or an overlapping promoter sequence [1,3]. TCPs can form homodimers or heterodimers from the same or other classes. However, heterodimers between specific members of the same class are more predominant. These TCP dimers reportedly show higher binding efficiencies [1].

Class I *TCP* genes generally promote cell division and plant growth, whereas class II genes inhibit cell division and regulate differentiation [6,7]. The genetic analysis of *TCP* single and multiple mutants or transgenic plants with overexpression of *TCP* and its dominant repressor form (*TCP-SRDX*) revealed that *TCP* genes exert a crucial role in various developmental processes, including seed germination [8,9,10], hypocotyl growth [11,12,13], trichome initiation and development [14,15], root growth [16], leaf patterning and development [17,18,19,20,21,22,23,24,25], axillary bud outgrowth [26], axillary branching [27], flowering [28,29,30], circadian clock [31], plant hormone biosynthesis, and signaling pathways [3,32].

Several studies reported that the *TCP* genes also regulate the development of floral organs, including petals, stamens, filaments, gynoecium, and ovules [33,34,35,36,37,38,39]. *TCP15*, a class I gene, is expressed in stamen filaments. The down-regulation of *TCP15*-related class I *TCPs*, or expression of *TCP15*’s dominant repressor form, results in short stamens. TCP15 and its related class I TCPs modulate GA-dependent stamen filament elongation by directly targeting *SAUR63* family genes, while *KNAT1* directly regulates the expression of *TCP15* [36,40]. *TCP4*, an miR319-targeted *CIN*-like gene, inhibits petal and stamen development when expressed in an miR319a mutant background, indicating that a functional *TCP4* is critical for developing these organs [35]. However, how *TCP4* influences stamen development needs to be determined. The up-regulation of *OsTCP1/PCF5*, rice’s *TCP4* orthologue, has been reported to result in a premature anther opening phenotype via the JA-dependent pathway [41]. *TCP24*, another miR319-targeted *CIN*-like gene, has been found to inhibit secondary cell wall thickening in the anther endothecium. *TCP24* overexpression in Arabidopsis results in non-dehiscent anthers, leading to a male sterile phenotype [42].

Recently, miR319 and its target *TCPs* (*TCP2/3/4/10/24*) were shown to control early-anther development [43], but there is no information on other *TCPs*. *TCP5* is a non-miR319-targeted member of the same clade (*CIN*-like) that regulates petal growth and differentiation [33,34]. Our study discovered that *TCP5* is expressed in the early anthers. Overexpression of *TCP5* and its dominant repressor form, *TCP5-SRDX*, resulted in smaller anthers with fewer pollen sacs and pollen. In addition, *TCP5* overexpression, like *TCP24*, resulted in a non-dehiscent anther phenotype, implying that *TCP5* may have a redundant function with other *CIN*-like *TCPs* during anther development.

## 2. Results

### 2.1. Overexpression of TCP5 Resulted in Male Sterility

We first acquired a T-DNA mutant (SM_3_29639) from the Nottingham Arabidopsis Stock Center to examine the putative function of *TCP5* in the reproductive phase. The T-DNA mutant was null, as validated by the RT-PCR data (Appendix A), but neither male nor female organs displayed any apparent abnormalities. We then overexpressed *TCP5* in Arabidopsis using the *UBQUITIN 10* promoter [44]. The transgenic plants developed longer and slightly narrower rosette leaves than the wild type (Figure 1A,B). During the flowering stage, the transgenic plants exhibited a dwarf and slightly bushy phenotype (Figure 1C), with the apex of their inflorescence usually twisted and the flowers arranged in clusters (Figure 1C and Appendix A). These flowers produced few to no seeds. However, they produced more seeds when pollinated with wild-type pollen, suggesting that the transgenic plants’ sterility was primarily due to the male side.

We dissected the flowers of transgenic plants with severe phenotypes using a stereomicroscope. The transgenic flowers were slightly smaller than the wild type (Figure 1D,H). The sepals and petals were morphologically normal, but the stamens, which consisted of short filaments and small anthers, surrounded a gynoecium with fewer stigmatic papillae (Figure 1E,I). Notably, these anthers did not shed pollen during the anthesis stage (Figure 1F,J). However, they produced living pollen inside, as evidenced by Alexander staining (Figure 1G,K), indicating that these anthers could not dehisce.

We examined *TCP5* expression levels in several transgenic plants with varying degrees of sterility. We discovered that plants with a more severe phenotype had more *TCP5* transcripts, revealing a positive correlation between male sterility and *TCP5* expression levels (Appendix A).

### 2.2. TCP5 Overexpression Reduced Lignin Accumulation in the Anther Endothecium

We used phloroglucinol staining and investigated the lignin content in the anthers of wild-type and *TCP5*-overexpressing plants with a severe phenotype to determine the cellular basis of anther non-dehiscence (Figure 1C and L15 in Appendix A). After staining, a dark red hue appeared in the wild-type anther loci. However, this color was much reduced in the anthers of *TCP5*-overexpressing plants (Figure 2A,B), indicating a lower lignin level in these transgenic anthers than in the wild type. Sufficient lignin accumulation on the anther endothecium is essential for anther dehiscence during anthesis. We thus focused on this layer after anther clearing. Continuous fiber band formation on the endothecium in wild-type anthers was indicative of constant thickening of the secondary cell wall (Figure 2B,C). However, the *TCP5*-overexpressing anthers exhibited discontinuous fiber band formation, meaning that no fiber bands developed on specific endothecium-layer cells, particularly in the upper half of the anther (Figure 2F,G). A cross-section of the anther at anthesis further supported the difference in fiber band distribution on the endothecium between the wild type and the *TCP5*-overexpressing plant (Figure 2D,H). The wild type developed dense fiber bands on the endothecium layer. In contrast, the *TCP5*-overexpressing plant had a few fiber bands on the counterpart region, though we occasionally found dense fiber bands near the connective tissue.

These findings demonstrate that *TCP5* overexpression reduced lignin accumulation on the endothecium, resulting in a non-dehisced anther phenotype.

### 2.3. TCP5 Overexpression Resulted in Fewer Pollen Sacs and Pollen

*TCP5* overexpression transgenics had fewer pollen sacs in their anthers and a dehiscence defect (Figure 3). The wild-type anther had four pollen sacs. However, transgenic plants with a severe phenotype developed two or three pollen sac anthers, with an average of 2.5 pollen sacs per anther (Figure 4A). Consequently, the transgenic plants also exhibited reduced pollen numbers. On average, the wild-type anther contained 321 pollen grains, but *TCP5*-overexpressing plants generated only 107–274 pollen grains (Figure 4B).

The defect in pollen sac formation occurred before the meiosis stage. In stage 4 anthers [45], sporogenous cells appeared in all four corners of the wild type but only two or three in the *TCP5*-overexpressing plants (Figure 3A,B). While the anthers from the *TCP5*-overexpressing plants only had two or three loci at stages 6–8, the wild-type anther had four. Meiosis appeared to proceed normally in the anther loci of both wild-type and *TCP5*-overexpressing plants (Figure 3C–F). At stage 9, microspores developed in the anther loci, and the tapetum gradually degenerated (Figure 3G,H). At stages 11–12, the fiber band developed on the endothecium of the anthers of both wild-type and *TCP5*-overexpressing plants, but less in the latter (Figure 3I,J). During anthesis, at stage 13, the wild-type anthers with dense fiber bands on the endothecium opened to release pollen (Figure 3K). The anthers of *TCP5*-overexpressing plants with fewer fiber bands on the endothecium could not open despite the septum disappearing and the stomium partially breaking down, resulting in no pollen release from these anther loci (Figure 3L). This is consistent with our observations in Figure 2.

### 2.4. TCP5 Overexpression Affected the Expressions of Genes Involved in Secondary Cell Wall Formation and Other Key Anther Genes

We investigated the expression level of genes involved in secondary cell wall formation and several vital anther genes to comprehend the molecular mechanism underlying the defective anther phenotype in the *TCP5*-overexpressing plants.

The secondary cell wall predominantly comprises cellulose, hemicellulose, and lignin. We used qRT-PCR to determine the expression levels of genes involved in cellulose biosynthesis (*IRX1*, *IRX3*, *IRX5*, *CESA1*, *IRX8*, and *IRX9*) [46,47,48,49] and lignin biosynthesis (*PAL4*, *C4H*, C3H, *4CL1*, *COMT1*, *CCOAMT*, *HCT*, *CAD4*, *CAD5*, *CAD6*, *F5H1*, and *CCR1*) [50]. The overexpression of *TCP5* in plants resulted in the down-regulation of all genes except *IRX1* and *IRX5*, which remained unchanged, and *CCOAMT*, which showed increased expression (Figure 5A). In addition, three transcription factor genes, *MYB26* and its downstream targets *NST1* and *NST2*, which influence secondary cell wall thickness [51,52,53,54], were down-regulated. *AHP4*, a negative regulator of secondary cell wall thickening [55], was significantly up-regulated in these *TCP5*-overexpressing plants compared to the wild type (Figure 5A).

Several genes are known to be activated during anther development. Twenty genes were selected to compare their expression levels in wild-type and *TCP5*-overexpressing plants. Except for *TPD1*, a tapetum-determinant gene [56], which showed no significant expression change, all nineteen genes were down-regulated in *TCP5* overexpression transgenics (Figure 5B). The most down-regulated genes were *BAM1* and *BAM2*, two receptor kinase-like genes involved in cell fate specification [57], and *AMS* and *DYT1*, two bHLH transcription factors which regulate tapetum development [58,59]. The second group consisted of two glutaredoxin genes, *ROXY1* and *ROXY2*, and two TF genes, *SPL9* and *TGA9* [60,61,62]. The third group included *SPL8* and *SPL2*, as well as *SERK1* and *EMS1*, which control anther sporogenesis, *SPL/NZZ*, a well-studied early-anther gene, and *BIM1*, which is involved in brassinosteroid signaling [61,63,64,65,66,67]. Next, we observed *MYB33* and *TGA10* [62,68], whose expression levels were lowered by one-third compared to the wild type. *SPL11*, *MYB65*, and *SERK2*, the final group of genes [61,65,68], had their expression levels reduced by two-thirds, slightly more than half, and half, respectively, compared to the wild type (Figure 5B).

These findings indicate that overexpressing *TCP5* can influence the expression of numerous genes involved in secondary cell wall thickening and early-anther development.

### 2.5. TCP5 Was Expressed in Early Anthers

We conducted RT-PCR analysis in various organs and flower buds or flowers at different developmental stages to investigate the *TCP5* expression profile. The highest expression signal was detected in the cauline leaf, followed by a moderate expression level in the seedling, rosette leaf, and inflorescence, a low expression signal in the young silique, a feeble signal in the stem and older silique, and no signal in the root (Appendix A). Furthermore, the *TCP5* expression signal was observed in all flower buds and open flowers, with the expression level slightly lower in open flowers (Appendix A). Contrarily, a robust GUS signal was discovered in the open flowers, including the sepal, petal, stamen, and upper part of the gynoecium, a weak signal in the young flower buds, and no signal in the early anthers (Appendix A).

Consequently, we performed RNA in situ hybridization for *TCP5* in wild-type anthers. At stages 2–3, the *TCP5* expression signal was evenly distributed throughout the region, with a strong signal in the corners (Figure 6A,B). Stages 4–5 saw the formation of the four lobes on the anthers. The *TCP5* signal was detected in the epidermis, endothecium, middle layer, tapetum, and sporogenous cells (or pollen mother cells/PMCs), as well as vascular bundles in the connective region (Figure 6C,D). At stages 6–8, the anthers underwent meiosis, and a strong *TCP5* signal appeared in the tapetum, PMCs, or tetrads, and the vascular bundles (Figure 6E–G). Following meiosis, the *TCP5* signal disappeared from stage 9 to stage 13 anthers (Figure 6H,I). Thus, we concluded that *TCP5* was expressed in the early anthers, contradicting the expression pattern observed in GUS plants (Appendix A).

### 2.6. TCP5-SRDX Transgenic Plants Also Had Male Fertility Defects

We used *UBQUITIN 10* and *SPL8* promoters to create *TCP5-SRDX* transgenic plants to rule out putative functional redundancy with other *TCPs*. *SRDX* is a sequence that encodes an EAR-motif-based artificial repression domain (LDLDLELRLGFA), which provides the fused TF with dominant transcriptional repression activity [69]. Both *pUBQ10:TCP5-SRDX* and *pSPL8:TCP5-SRDX* transgenic plants appeared to be normal, except for the absence of elongated siliques after flowering, at least for the early-arising flowers. Their flower size was comparable to the wild type (Figure 7A,D,G). However, their anthers were noticeably smaller and produced less pollen than the wild type (Figure 7B,C,E,F,H,I). This phenotype appeared to be substantially more severe in the *pUBQ10:TCP5-SRDX* transgenic plants. Interestingly, these anthers could open to release even less pollen at anthesis (Figure 7F,I). Cross-sections and Alexander staining of these anthers revealed fewer pollen sacs and little or no pollen (Appendix A and Figure 7J,K,L). Phloroglucinol staining showed that the dark red color of the lignin in the majority of these anthers was comparable to the wild type (Figure 7M–O), indicating the ability of these anthers to dehisce.

### 2.7. TCP5-SRDX Transgenic Plants Exhibited Altered Expression of Early-Anther Genes

The expression of *TCP5-SRDX* in Arabidopsis resulted in undersized anthers with fewer pollen sacs and less pollen (Figure 7 and Appendix A). Thus, we compared the expression levels of the twenty early-anther genes (Figure 5B) in wild-type and *TCP5-SRDX* transgenic plants. All genes were significantly down-regulated in the *pUBQ10:TCP5-SRDX* transgenics, except *SPL2*, which showed no discernible change in expression level (Figure 8A). *AMS* was the most down-regulated gene, followed by *BAM1/2*, *MYB33*, *BIM1*, *TGA9/10*, and *SPL/NZZ*. *TPD1*, *SERK1/2*, *ROXY2*, *SPL9*, *SPL11*, and *MYB65* were next, while the rest, including *EMS1*, *DYT1*, *SPL8*, and *ROXY1*, were less down-regulated, with expression levels not lowered by more than 50% compared to the wild type (Figure 8A). Most of these genes were also down-regulated in the *pSPL8:TCP5-SRDX* transgenic plants. *AMS* was similarly the most down-regulated gene, with nearly undetectable expression (Figure 8B). *TGA9* and *DYT1*, the second most down-regulated genes, had their expression levels drop below 10%. However, unlike *pUBQ10:TCP5-SRDX* transgenics, *ROXY1* and *SPL11* expression levels did not alter significantly, although *SPL9* expression levels increased dramatically (Figure 8).

Nevertheless, these results demonstrated that most early-anther genes were down-regulated in *TCP5-SRDX* transgenic plants.

## 3. Discussion

Plant fertility refers to the male and female activities in the flower and is affected by various factors in the reproductive process. The male reproductive organ, the stamen, located in the flower’s third whorl, consists of an anther and a filament. The anther, a butterfly-like structure which produces, stores, and releases pollen, is located at the tip of the filament, a slender stalk which connects the anther to the floral central axis [70]. Any developmental changes or defects in these structures may impact male fertility, which involves numerous genes or regulatory networks.

Our study discovered that *TCP5* is involved in anther development. *TCP5*-like genes (*TCP5*, *TCP13*, and *TCP17*) may have functional redundancy, because the single knock-out mutant *tcp5* did not exhibit an apparent phenotype in anther development as expected. Thus, the triple mutant *tcp5 tcp13 tcp17* requires further investigation. Overexpressing *TCP5* in Arabidopsis resulted in a non-dehiscent anther phenotype at anthesis (Figure 1). Further investigation revealed that the endothecium in the anther of the *TCP5*-overexpressing plant exhibited decreased lignin accumulation and fibrous band formation (Figure 2), which could account for the anther’s non-dehiscence. *TCP24*, an miR319-targeted gene, demonstrated a similar anther phenotype when overexpressed [42]. However, it appeared more severe than the *TCP5*-overexpressing plants because of the total absence of lignin accumulation in the anther endothecium [42]. *TCP5* and *TCP24* were both expressed in early anthers, including in the epidermis, endothecium, middle layer, tapetum, and PMCs, but not in late anthers in wild-type plants. The endothecium of late anthers did not exhibit any expression of these two genes during secondary cell wall thickening. Thus, we conclude that *TCP5* and *TCP24* act as negative regulators of secondary cell wall thickening, preventing precocious secondary cell wall thickening in the early-anther endothecium. Intriguingly, the *TCP24* signal was not detected in the endothecium layer of the late anthers in the *TCP24*-overexpressing plants. This indicates that the non-dehiscent anther phenotype resulted from enhanced *TCP24* expression rather than ectopic expression [42]. Thus, we need to examine the detailed expression profile of *TCP5* in *TCP5*-overexpressing plants. *TCP5* and *TCP24* are both members of the CIN-clade, although they belong to separate subclades (*TCP5*-like and Jaw-D/miR319-targeted), suggesting that other *CIN*-like genes may perform similar functions. Many genes related to cellulose (*IRX1/3/8/9* and *CESA1*) and lignin biosynthesis (*PAL4*, *C3H*, *C4H*, *4CL1*, *COMT1*, *CAD4/5/6*, *HCT*, *F5H1*, and *CCR1*) were down-regulated in *TCP5*-overexpressing plants, which supported the existence of their non-dehiscent anther phenotype (Figure 5). A previous study also showed the down-regulation of *IRX1/3/5*, *C4H*, *4CL1*, *CCOAMT*, and *PAL4* genes in *TCP24*-overexpressing plants [42]. Both *TCP5* and *TCP24*-overexpressing plants showed increased levels of *AHP4* (*Arabidopsis Histidine-containing Phosphotransfer Factor 4*), which is consistent with a prior finding that overexpression of *AHP4* results in a non-dehiscent anther phenotype [55]. *MYB26* and its two targets *NST1* and *NST2*, three transcription factors that regulate secondary cell wall thickening, were down-regulated in *TCP5*-overexpressing plants, but *MYB26* exhibited no significant change in *TCP24*-overexpressing plants [42]. Several other discrepancies exist between these two transgenic plants. In the *TCP5*-overexpressing plants, *CCOAMT* was up-regulated, while *IRX5* did not exhibit any significant expression change. In contrast, these two genes were down-regulated in the *TCP24*-overexpressing plants. These expression differences for some of the genes mentioned above indicate that *TCP5* and *TCP24* somewhat differ in regulating secondary cell wall thickening when overexpressed.

Another distinguishing feature of the *TCP5*-overexpressing plant was a slightly smaller anther with fewer pollen sacs and pollen (Figure 3 and Figure 4). The *TCP5-SRDX* transgenic plants, where *TCP5* functioned as a transcriptional repressor, remarkably exhibited a similar anther phenotype, except for anther dehiscence (Appendix A). The lignin intensity in the anthers of *TCP5-SRDX* plants was comparable to that of the wild type, as indicated by a dark red hue from phloroglucinol staining (Figure 7). This differed from *TCP24*, in which the *TCP24-SRDX* plants accumulated more lignin in the anthers than the wild-type plants [42]. It was unclear whether the *TCP24* transgenic plants produced fewer pollen sacs and pollen in the anthers. Nonetheless, a recent study reported that miR319 and its target genes *TCP2/3/4/10/24* contribute to early-anther development. Multiple mutants of these genes exhibited a similar early-anther phenotype to *TCP5*-overexpressing and *TCP5-SRDX* plants [43]. These findings demonstrated that *CIN-like TCP* genes are involved in early-anther development.

Other genes, such as *SPL/NZZ*, *ROXY1/ROXY2*, *TGA9/TGA10*, and *SPL8/*miR156-*SPL*, have previously been shown to control the early-anther development, affecting pollen sac formation and pollen production [60,61,62,64,67]. Single and/or multiple mutants of these genes resulted in fewer pollen sacs or even pollen sac anthers, such as those from *TCP5*-overexpressing/*TCP5-SRDX* plants and *mir319a/b/c* mutants [43]. SPL/NZZ has been shown to recruit TPL (TOPLESS) and interact with CIN-like TCPs, including TCP5, to repress the expression of these *TCP* genes [71,72]. *TCP4*, miR319’s target, directly binds to the promoters of *TGA9/TGA10* and *ROXY2*, repressing their expression [43]. Many early-anther genes, including *SPL/NZZ*, *BAM1/BAM2*, *AMS*, *ROXY1/ROXY2*, *TGA9/TGA10*, *SPL8*, and others [57,58,60,61,62,64,67], were significantly down-regulated in both *TCP5*-overexpressing and *TCP5-SRDX* plants (Figure 8), indicating that a bona fide *TCP5* expression is critical for early-anther development. However, it is unknown whether TCP5 can bind to the cis-elements of these genes to regulate their expression or interact with them to govern early-anther development.

## 4. Materials and Methods

### 4.1. Plant Materials and Growth Conditions

*Arabidopsis thaliana* ecotype Columbia-0 (Col-0) was used as the wild type. All other plants used in this study were developed in the Col-0 background. A *tcp5* mutant (SM_3_29639) was acquired from the Nottingham Arabidopsis Stock Centre (NASC). Seeds were sown in a plastic Petri dish lined with wet tissue paper and stratified for two days in the dark at 4 °C. The stratified seeds were transferred into pots filled with a mixed substrate of nutrient soil (non-sterilized without fertilizer)–vermiculite (1:1). They were grown in a greenhouse with light intensity 120 μmol/m^2^/s at 22–23 °C and 60–70% relative humidity under long-day conditions (16 h light/8 h dark).

### 4.2. Plasmid Construction and Transgenic Plants

The *pUBQ10:TCP5* and *pUBQ10:TCP5-SRDX* constructs were created using gateway cloning. The full-length coding sequence (CDS) of *TCP5* was amplified by PCR from wild-type Arabidopsis inflorescence cDNA, and the *TCP5-SRDX* sequence (*TCP5*-CDS without the stop codon fused with CTTGATCTTGATCTTGAACTTAGACTTGGATTTGCTTAA, a sequence encoding the EAR motif with a stop codon) was synthesized directly by Sangon Biotech, Shanghai, China. These sequences were added to pDONR207 separately using BP reactions (Gateway™ BP Clonase™ II enzyme mix, Invitrogen/Thermo Fisher Scientific, Shanghai, China) and then transferred into the destination vector pUB-Dest [44] using LR reactions (Gateway™ LR Clonase™ II enzyme mix, Invitrogen/Thermo Fisher Scientific, Shanghai, China). A *pOLE:OLE-TagRFP* cassette expressed in seeds from a pDe-CAS9 derivative [73,74,75] was cut off and integrated into *pUBQ10:TCP5* and *pUBQ10:TCP5-SRDX*, respectively, to allow for visual screening of the transgenic plants. The 2.1 kb *SPL8* promoter, expressed in early anthers [61], was acquired via PCR from wild-type leaf genomic DNA for the synthesis of *pSPL8:TCP5-SRDX*, and the *pUBQ10* promoter in *pUBQ10:TCP5-SRDX* was replaced through appropriate restriction enzyme digestion and ligation. A 5173 bp *TCP5* promoter was amplified on the wild-type genomic DNA for *pTCP5:GUS* construction, and it was inserted into the pGPTV-BAR [76] just before the β-glucuronidase (*uidA*) reporter gene. PCR, restriction enzyme digestion, and sequencing validated the plasmids. These plasmids were then transformed into wild-type plants by floral dipping using *Agrobacterium tumefaciens* strain GV3101 [77]. Basta spraying or screening red seeds with a fluorescence stereomicroscope helped in the selection of transgenic plants (Leica M205C; Leica Microsystems, Mannheim, Germany). Appendix A includes a list of the primers used. More information regarding plasmid construction and transformation is available in the Appendix A.

### 4.3. Histology, Histochemistry, and Microscopy

Unopened anthers were dissected from flowers just before anthesis and submerged in drops of Alexander staining solution [78] on a slide for 30–60 min at room temperature. The slide was then covered with a coverslip and examined under a bright-field microscope (Leica DFC450; Leica Microsystems, Mannheim, Germany).

Phloroglucinol staining was used to visualize lignin deposition in the anthers, as previously described [42]. We removed the sepals and/or petals from open or unopened flowers, dyed the rest of the flowers for 5–10 min, and examined them using a stereomicroscope with a bright field (Leica M205C; Leica Microsystems, Mannheim, Germany).

The inflorescences or open flowers of 4- or 5-week-old plants were fixed in 4% glutaraldehyde for at least 24 h at 4 °C to create cross-sections of the anthers, which were then embedded in paraffin following dehydration. A microtome (RM2235, Leica Biosystems Nussloch GmbH, Nussloch, Germany) was used to cut 5 μm thick sections, which were then deparaffinized, hydrated, and stained with 0.05% (*w*/*v*) toluidine blue for 1–3 min at room temperature. The stained slides were rinsed three times with water, dried at room temperature, and sealed with neutral balsam (Beijing Solarbio Science & Technology Co., Ltd., Beijing, China) under a fume hood.

The anthers from stages 12–13 were dissected and cleared for 2–4 h in a chloral hydrate–glycerol–water (8:1:2/*w*:*v*:*v*) clearing solution to see the distribution of fibrous bands on the endothecium of the intact anthers. The anthers were then observed under a microscope (Leica DFC450) with differential interference contrast optics.

Individual unopened anthers were stained in the Alexander staining solution before being smashed and photographed to measure the pollen output. We used Image J (version 1.54e) to count the pollen [79].

### 4.4. In Situ Hybridization

RNA in situ hybridization was performed using the previously reported method [80]. The full-length CDS of *TCP5* was amplified by PCR using forward and reverse gene-specific primers containing a T3 sequence (aattaaccctcactaaaggg, a consensus promoter sequence for T3 polymerase) and a T7 sequence (gtaatacgactcactatagggc, a consensus promoter sequence for T7 polymerase), respectively. The PCR product was gel-purified and quantified with a NanoDrop 2000 spectrophotometer (Thermo Fisher Scientific, Waltham, MA, USA). T7 RNA polymerase (Roche Diagnostics GmbH, Mannheim, Germany) was used to synthesize the *TCP5* antisense probe per the manufacturer’s instructions. The synthesized *TCP5* probe was hydrolyzed before use. Probe hybridization was conducted at 50 °C in a drying oven (GZX-9246MBE, BOXUN, Shanghai, China) overnight.

### 4.5. RNA Extraction and qRT-PCR Analysis

Total RNA from the inflorescences was extracted using RNAiso Plus reagent (Takara, Japan) and then processed with DNase I (Takara Bio Inc., Beijing, China) to remove DNA contamination. RNA integrity was assessed by running a 1.2% denaturing agarose gel. RNA quality and concentration were further determined using a NanoDrop 2000 spectrophotometer. Following the manufacturer’s instructions, 1 μg of total RNA was used to synthesize first-strand cDNA using an RNA HiScript^®^ III RT SuperMix for qPCR (+gDNA wiper) Kit (Vazyme, Nanjing, China). In parallel, a no-reverse transcription control was included using the *RAN3* primer pair (5′-ACCAGCAAACCGTGGATTACCCTAGC-3′ and 5′-ATTCCACAAAGTGAAGATTAGCGTCC-3′) to evaluate the purity of the RNA sample.

We used Roche-LightCycler-480 II Real-Time System with ChamQTM Universal SYBR qPCR Master Mix (Promega, Beijing, China) to conduct qRT-PCR. The qRT-PCR reaction volume was 20 μL, with 10 μL of 2× SYBR qPCR Master Mix, 0.4 μL of primers (10 μM), 8 μL of cDNA template (50 ng/μL), and 1.2 μL of ddH_2_O. The reaction was performed at 94 °C for 2 min, followed by 40 cycles of 94 °C for 15 s, and 60 °C for 31 s. Here, *18S rRNA* served as an internal control. The experiments included three technical and three biological replicates. Gene expression levels were determined using the 2^−^^∆∆CT^ method. Appendix A lists the gene-specific primers used for qRT-PCR.

### 4.6. Data Analysis

The genes’ relative expression levels were estimated using a one-way analysis of variance (ANOVA) and post hoc Tukey’s test. The rest of the data were evaluated using the Student *t*-test. All data are presented as the mean ± SD. The asterisks indicate significant differences between the genes or groups (**, *p ≤* 0.01; ***, *p* ≤ 0.001; and ****, *p* ≤ 0.0001).

## 5. Conclusions

As plant-specific transcriptional regulators, TCP family members regulate multiple developmental processes during a plant’s life cycle. *TCP5* belongs to the CIN clade but is not targeted by miR319. Our study discovered that *TCP5* is expressed in early anthers. The overexpression of *TCP5* or its dominant repressor form *TCP5-SRDX* resulted in undersized anthers with fewer pollen sacs and pollen, as well as a non-dehiscent anther phenotype in *TCP5*-overexpressing plants. Consistent with their phenotype, several early-anther genes and genes associated with secondary cell wall thickening changed their expression levels in these transgenic plants. Our findings demonstrated that adequate *TCP5* expression is essential for early-anther development, as miR319-targeted *CIN* genes have previously been shown to regulate early-anther development [43]. Thus, all *CIN* genes in Arabidopsis might play a role in early-anther development. The relationship between these *TCPs* and other early-anther genes, such as *SPL8* and miR156-targeted *SPLs*, should be explored further.

## Figures and Tables

**Figure 1 ijms-26-01813-f001:**
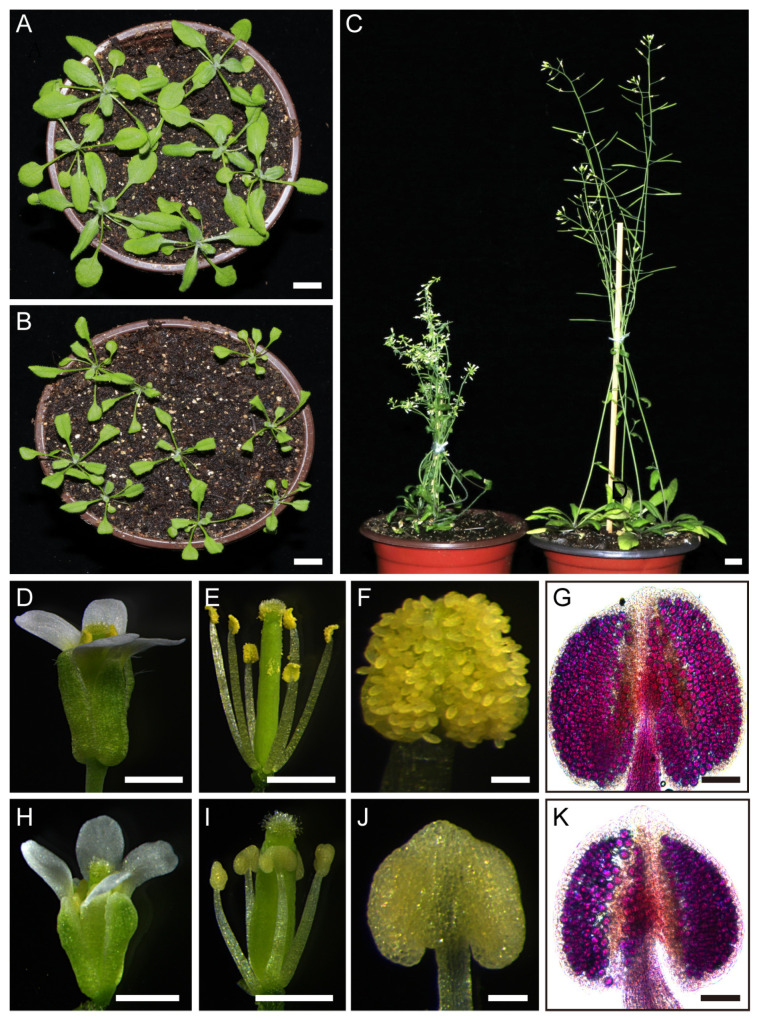
Phenotypes of *pUBQ10:TCP5* transgenic plants: (**A**) three-week-old wild-type plants; (**B**) three-week-old *pUBQ10:TCP5* plants with slightly narrower downward rosette leaves; (**C**) six-week-old *pUBQ10:TCP5* plants (left) showing a dwarf and non-elongated silique phenotype compared with the wild type (right); (**D**–**K**) flower and anther morphology; (**D**,**H**) flowers of the wild type (**D**) and the *pUBQ10:TCP5* plant (**H**) at the anthesis stage; (**E**,**I**) sepals and petals removed from the wild type (**E**) and the *pUBQ10:TCP5* flowers (**I**); (**F**,**J**) wild-type anther with many released pollen grains (**F**) and a non-dehisced anther from the *pUBQ10:TCP5* plant (**J**) at anthesis; and (**G**,**K**) wild-type anther (**G**) and *pUBQ10:TCP5* anther (**K**) after Alexander staining. Scale bars: 1 cm in (**A**–**C**); 1 mm in (**D**,**E**,**H**,**I**); and 50 μm in (**F**,**G**,**J**,**K**).

**Figure 2 ijms-26-01813-f002:**
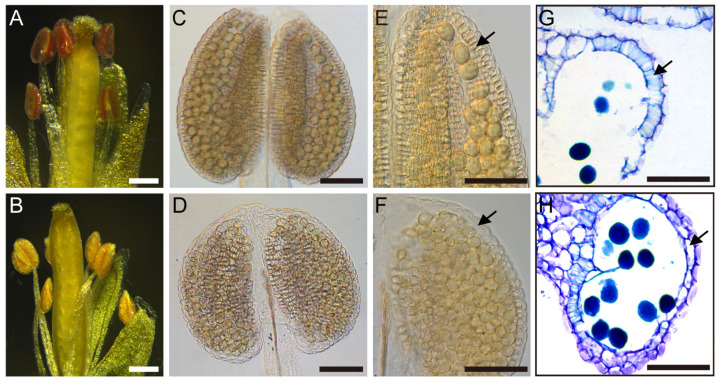
Secondary cell wall thickening of the anthers: (**A**,**B**) phloroglucinol staining of wild-type (**A**) and *pUBQ10:TCP5* (**B**) flowers (petals removed), with the lignified materials in a dark red color; (**C**,**D**) cleared wild-type (**C**) and *pUBQ10:TCP5* (**D**) whole anthers; (**E**,**F**) magnified parts of anthers in (**C**,**D**), noting the difference in fibrous band distribution on the endothecium (arrows); and (**G**,**H**) cross-sections of wild-type (**G**) and *pUBQ10:TCP5* anthers (**H**) at anthesis, with the fibrous bands on endothecium stained in a light blue color. Arrows indicate the endothecium. Scale bars: 250 μm in (**A**,**B**) and 50 μm in (**C**–**H**).

**Figure 3 ijms-26-01813-f003:**
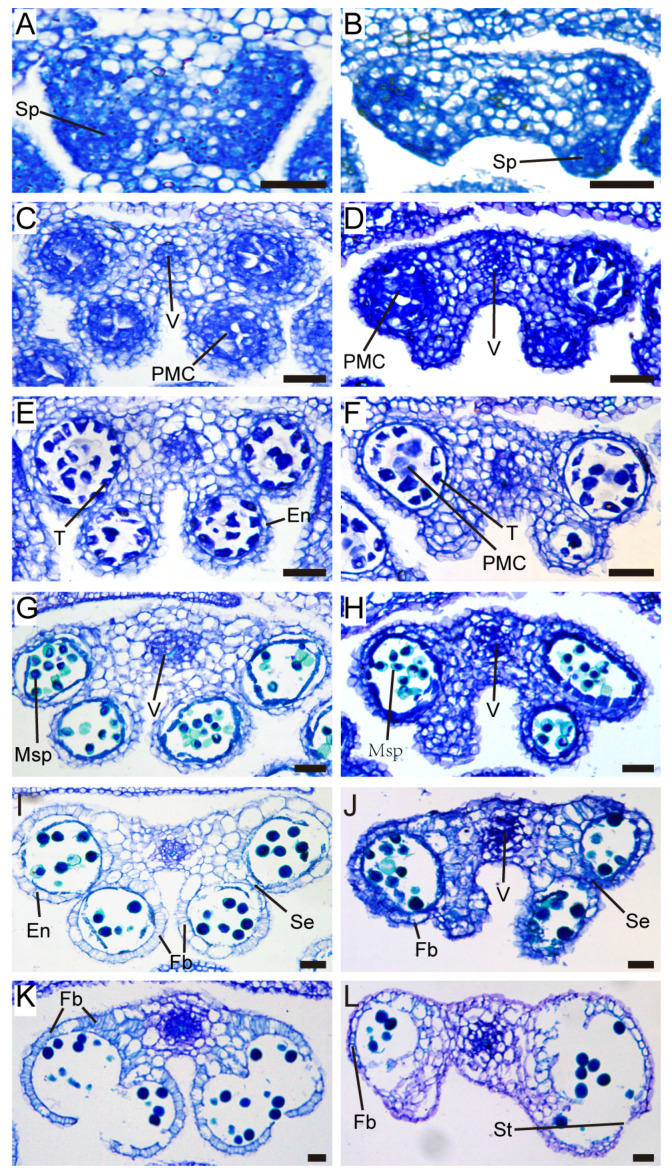
Defects of the *pUBQ10:TCP5* anthers: (**A**,**B**) cross-sections of stage 4 wild-type (**A**) and *pUBQ10:TCP5* (**B**) anthers, with the sporogenous cells stained in a dark blue color; (**C**–**F**) cross-sections of wild-type (**C**,**E**) and *pUBQ10:TCP5* (**D**,**F**) anthers at the meiosis stages; (**G**,**H**) cross-sections of wild-type (**G**) and *pUBQ10:TCP5* transgenic anthers (**H**) at stage 9; (**I**,**J**) cross-sections of wild-type (**I**) and *pUBQ10:TCP5* transgenic anthers (**J**) at stage 11, with fibrous bands appearing on the endothecium; and (**K**,**L**) cross-sections of stage 13 anthers of wild-type (**K**) and *pUBQ10:TCP5* transgenic plants (**L**). En, endothecium; Fb, fibrous band; Msp, microspore; PMC, pollen mother cell; Se, septum; Sp, sporogenous cell; St, stomium; T, tapetum; and V, vascular bundle. Scale bars: 25 μm in (**A**–**L**).

**Figure 4 ijms-26-01813-f004:**
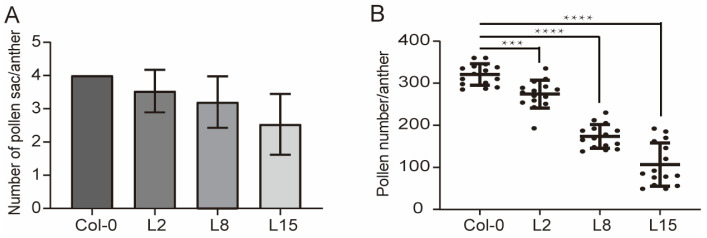
The number of pollen sacs and pollen in *pUBQ10:TCP5* plants: (**A**) number of pollen sacs from three different *pUBQ10:TCP5* transgenic plants compared to the wild type, with the pollen sac number counted from fifteen anthers (cross-sections of anthers from five plants); and (**B**) pollen number per anther in the wild type and three different *pUBQ10:TCP5* plants. ***, *p* ≤ 0.001; ****, *p ≤* 0.0001.

**Figure 5 ijms-26-01813-f005:**
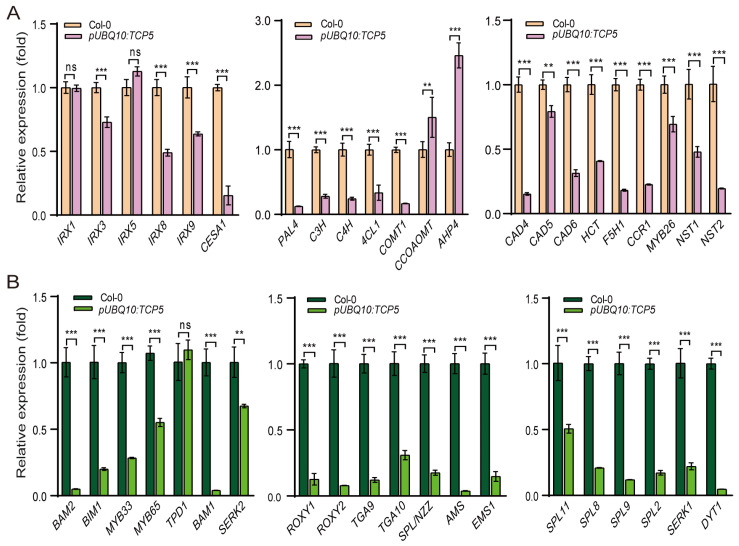
Overexpression of *TCP5* affected the expression profiles of many genes involved in secondary cell wall thickening and early-anther development: (**A**) expression levels of genes associated with secondary cell wall thickening in *pUBQ10:TCP5* plants; and (**B**) expression levels of genes involved in early-anther development in *pUBQ10:TCP5* plants. ns, not significant; **, *p* ≤ 0.01; and ***, *p* ≤ 0.001.

**Figure 6 ijms-26-01813-f006:**
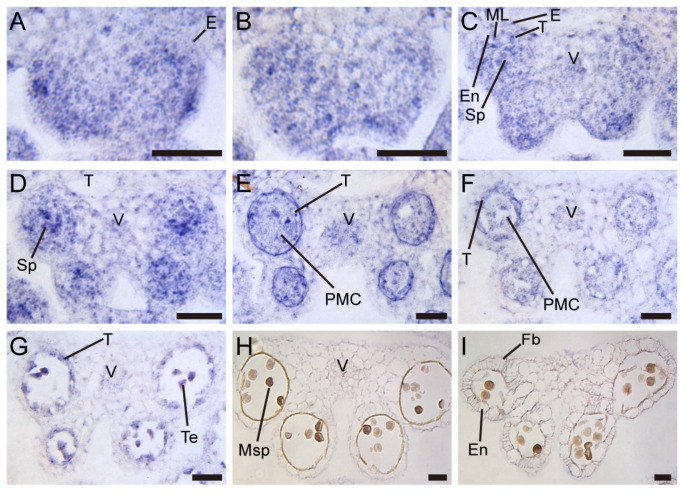
Spatiotemporal expression of *TCP5* in wild-type anthers detected by RNA in situ hybridization: (**A**–**D**) before meiosis, the transverse sections of a stage 2 anther (**A**), a stage 3 anther (**B**), a stage 4 anther (**C**), and a stage 5 anther (**D**); (**E**–**G**) meiosis, the transverse sections of anthers at stage 6 (**E**), stage 7 (**F**), and stage 8 (**G**); and (**H**,**I**) after meiosis, the transverse sections of anthers at stage 10 (**H**) and stages between 11 and 12 (**I**). E, epidermis; En, endothecium; Fb, fibrous band; ML, middle layer; Msp, microspore; PMC, pollen mother cell; Sp, sporogenous cell; T, tapetum; Te, tetrad; and V, vascular bundle. Scale bars: 25 μm in (**A**–**I**).

**Figure 7 ijms-26-01813-f007:**
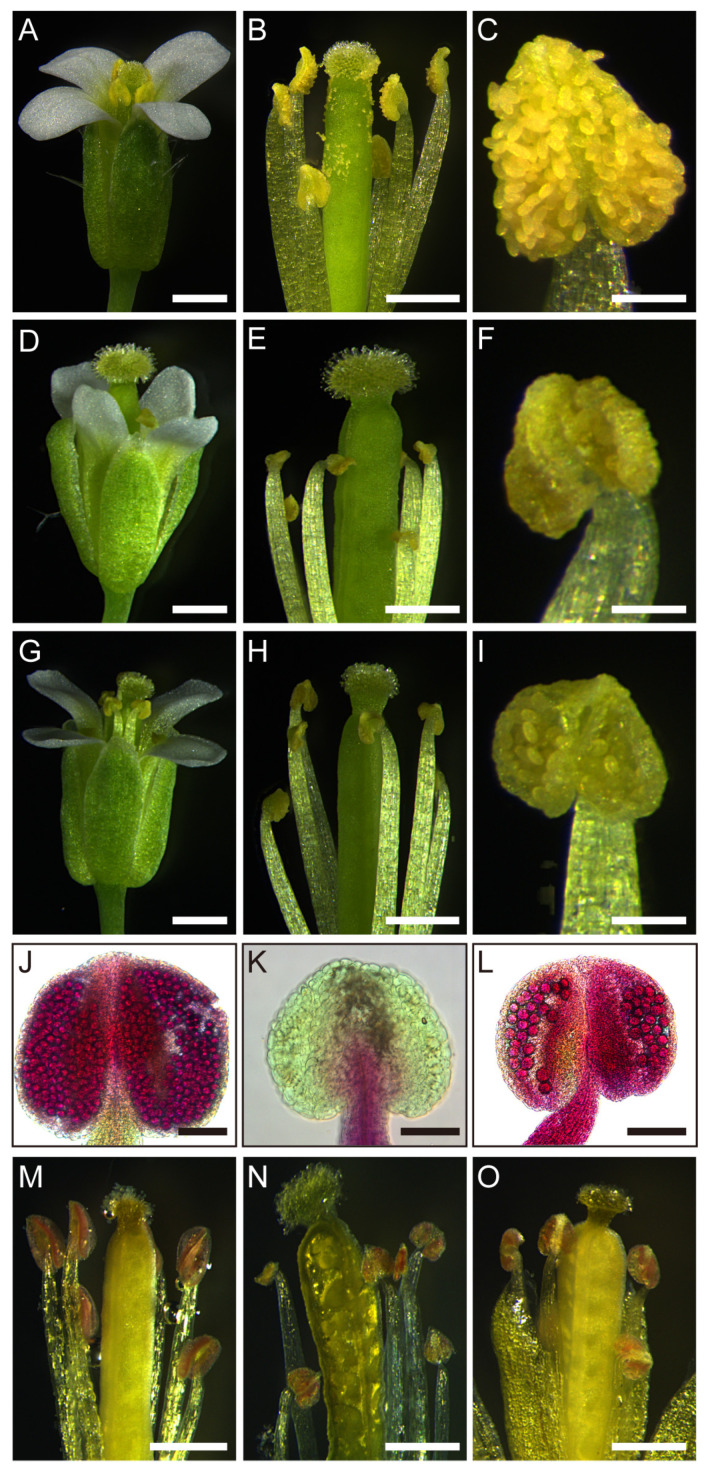
Flower and anther phenotypes of *TCP5-SRDX* transgenic plants: (**A**) an intact wild-type flower; (**B**) a wild flower with all sepals and petals removed; (**C**) a wild-type anther at anthesis; (**D**–**F**) an intact flower (**D**), a flower with all sepals and petals removed, (**E**) and an anther at anthesis (**F**) from a *pUBQ10:TCP5-SRDX* plant; (**G**–**I**) an intact flower (**G**), a flower with all sepals and petals removed (**H**), and an anther at anthesis (**I**) from a *pSPL8:TCP5-SRDX* plant; (**J**–**L**) wild-type (**J**), *pUBQ10:TCP5-SRDX* (**K**), and *pSPL8:TCP5-SRDX* (**L**) anthers after Alexander staining; and (**M**–**O**) phloroglucinol staining of the flowers with some sepals and all petals removed from wild-type (**M**), *pUBQ10:TCP5-SRDX* (**N**), and *pSPL8:TCP5-SRDX* (**O**) plants. Scale bars: 500 μm in (**A**,**D**,**G**); 250 μm in (**B**,**E**,**H**,**M**–**O**); and 50 μm in (**C**,**F**,**I**,**J**–**L**).

**Figure 8 ijms-26-01813-f008:**
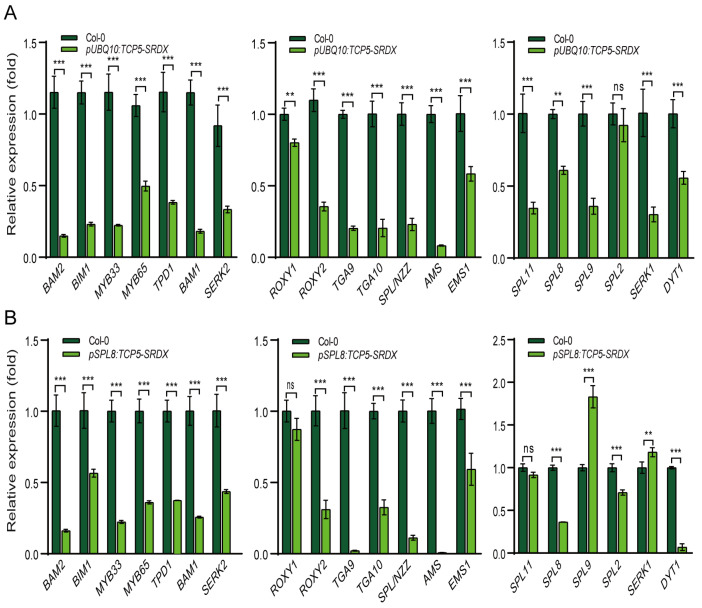
Expression profiles of early-anther genes in *TCP5-SRDX* transgenic plants: (**A**) expression changes in twenty early-anther genes in *pUBQ10:TCP5-SRDX* plants compared with the wild type; and (**B**) expression changes in the twenty early-anther genes in *pSPL8:TCP5-SRDX* plants compared with the wild type. ns, not significant; **, *p* ≤ 0.01; and ***, *p* ≤ 0.001.

## Data Availability

The data that support the findings of this study are all included in this paper.

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
