# Peer review of "Overexpression of TCP5 or Its Dominant Repressor Form, TCP5-SRDX, Causes Male Infertility in Arabidopsis"

_ijms, 2025, doi:10.3390/ijms26051813_

Round 1

Reviewer 1 Report

Comments and Suggestions for Authors

The manuscript is devoted to the study of overexpression of TCP5 or Its Dominant Repressor Form, TCP5-SRDX, and the study of anther phenotype, which has no basis for the analysis of the effect of TCP5 overexpression on anther growth and development.

An important component of the regulation of anther growth and development is their hormonal control, which the authors mention only briefly about jasmonic acid.

In my opinion, the manuscript should be substantially revised, in terms of the introduction, discussion of the results and in the conclusion

Author Response

Please see the attached pdf file.

Reviewer 2 Report

Comments and Suggestions for Authors

Dear Editor,

I am pleased to have reviewed the manuscript "Overexpression of TCP5 or Its Dominant Repressor Form, TCP5-SRDX, Causes Male Infertility in Arabidopsis." The manuscript is interesting and relevant to a broad audience concerned with stress responses. Gene expression analysis is always a field of interest due to the valuable information it provides, especially as it represents an invaluable resource for genetic improvement.

Abstract

The abstract is coherently written; however, I suggest adding a paragraph at the end summarizing the contribution of this research to the field.

Introduction

The authors provide an introduction that contextualizes the study; however, there are some errors that should be corrected:

  1. Scientific names must follow proper citation rules.
  2. The introduction mentions various genes and their effects on plants. While this information is important, it is difficult to follow. It is not necessary, but consider including a Table summarizing this information. Again, this is optional but worth considering if time permits.

Materials and Methods

The methodology presented is detailed but has some omissions, inconsistencies, and areas that could benefit from greater clarity or justification. Here are some points for improvement:

  1. Standardize the use of "°C," as sometimes it is written adjacent to the figure and sometimes separated.
  2. The relative humidity in the greenhouse environment is not mentioned, which is important for the growth of Arabidopsis thaliana.
  3. It is not specified whether the substrate was sterilized or contained fertilizers.
  4. Line 389: While the sequence used for fusion is mentioned, it would be prudent to explain its purpose rather than just mentioning it.
  5. The enzyme system used for BP and LR reactions is not mentioned.
  6. The cloning steps could include more technical details, such as DNA concentration or exact reaction conditions.
  7. Information is missing in the plasmid construction section. While details about vector construction are provided (it would be helpful to include NCBI accession numbers for the vectors used), the transformation with Agrobacterium lacks details such as the optical density used, transformation time, culture media, etc.
  8. The description of staining solution times and concentrations (e.g., Alexander and phloroglucinol) is ambiguous.
  9. The type of mounting or sealing used after neutral balsam staining is not specified.
  10. It is not mentioned whether positive or negative control probes were included to validate the in situ hybridization results.
  11. The temperature and duration of probe hybridization are not mentioned.
  12. The method used to evaluate RNA integrity (e.g., agarose gel electrophoresis) is not specified.
  13. Time ranges like "5-10 min" are too broad and should be more precise.
  14. It is not indicated whether additional tests were conducted to confirm the absence of DNA contamination.
  15. Line 427: ImageJ is mentioned for measurements, but it requires a citation, including the version used. The same applies to all software mentioned.
  16. Line 432: The term "T7 and T3 sequences" should be clarified.
  17. Hâ‚‚O should have the "2" in subscript.

Results

The results are interesting, though some information needs clarification:

  1. Line 93: Does TCP5 refer to the gene or the protein? If it refers to the gene, it should be italicized.
  2. In Table S1, the purpose of all primers is not described. It would also be helpful to indicate which are the forward (Fw) and reverse (Rv) primers.
  3. Figure S3: The significance is not mentioned. Consider adding letters for mean comparison analysis to each bar.
  4. Line 202: Twenty genes were selected, but the criteria for their selection are not mentioned.
  5. The reduction in transcripts for each gene is described, but this is purely descriptive. It would be prudent to include data showing the real reduction to make this section more objective.
  6. This section should remain objective. In line 217, there is speculation, which is inappropriate for this section and should be moved to the Discussion.
  7. Terms like qRT-PCR, qPCR, and RT-PCR should be standardized throughout the document.

Discussion

The discussion is interesting and well-approached but would benefit from a more extensive comparison with other studies:

  1. Some sentences are overly long and complex, making them harder to follow.
  2. Although references are used to compare the results, this section lacks discussion with a broader range of reports.

Conclusions

No comments.

References

References should be standardized. The references are not uniform and need to be carefully reviewed. If the journal permits, it would be prudent to include DOIs for each reference.

Author Response

Please see the attached pdf file

Reviewer 3 Report

Comments and Suggestions for Authors

Review result made on manuscript ID: ijms- 3368637 entitled as “Li et al., Overexpression of TCP5 or Its Dominant Repressor Form, 2 TCP5-SRDX, Causes Male Infertility in Arabidopsis” with the objectives: to investigate TCP5 OX and repression impact anther development in relation to male infertility using genetic and molecular analysis.

The manuscript is well articulated with relevant data presentation and sound interpretation or results, discussion with conclusion. Please try to address the following limitation and concerns given below.

Comments/suggestions/questions to be addressed:  

Question-1: why the null mutant of TCP5 did not show phenotype? How do you justify the biological function of TCP5 in repression process phenotypic results according to the statement given in line #92-95?

Question-2: Why does the repression of TCP5 show similar phenotypes as OX?

Comment on figures’ legends letters representation: Letters, A, B, C …. etc given to define figures’ legends are not standardized. Except for a few figures like figure 1, 4, 5, the labeling and description of each letter is confusing. For instance, Figure 2 labels or letters difficult to understand which stand for what line#145-150. The same holds true for figures 3, 6, 7,

Thank you!

Author Response

Please see the attached pdf file

Round 2

Reviewer 2 Report

Comments and Suggestions for Authors

Dear Editor,

I have reviewed the revised version of the manuscript. The document has improved considerably, especially the methodological section. It is coherent, and the analyses are solid. The authors have taken my previous comments into account.